# The W.A.I.O.T. Definition of High-Grade and Low-Grade Peri-Prosthetic Joint Infection

**DOI:** 10.3390/jcm8050650

**Published:** 2019-05-10

**Authors:** Carlo Luca Romanò, Hazem Al Khawashki, Thami Benzakour, Svetlana Bozhkova, Hernán del Sel, Mahmoud Hafez, Ashok Johari, Guenter Lob, Hemant K Sharma, Hirouchi Tsuchiya, Lorenzo Drago

**Affiliations:** 1Studio Medico Cecca-Romanò, Corso Venezia, 20121 Milano, Italy; mail@romano.institute; 2Romano Institute, Rruga Ibrahim Rugova 1, 00100 Tirane, Albania; 3Advanced Medical Center, Riyadh 12482, Saudi Arabia; bonedoc57@gmail.com; 4Zerktouni Orthopaedic Clinic, Casablanca 20000, Morocco; t.benzakour@gmail.com; 5R.R. Vreden Russian Research Institute of Traumatology and Orthopaedics, 195427 S. Petersburg, Russia; clinpharm-rniito@yandex.ru; 6RNIITO Department of Prevention and Treatment of Wound Infection, 195427 S. Petersburg, Russia; 7Department of Orthopaedics, British Hospital Buenos Aires, Buenos Aires C1280, Argentina; hdelsel@argentina.com; 8Department of Orthopaedics, October 6 University, Cairo 12566, Egypt; mhafez@msn.com; 9Department of Paediatric Orthopaedics and Spine Surgery, Children’s Orthopedic Centre, Mumbai 400016, India; drashokjohari@hotmail.com; 10Section Injury Prevention, DGOU, 10117 Berlin, Germany; prof.lob@medlob.de; 11Hull University Teaching Hospitals, Anlaby Road, Hull HU3 2JZ, UK; hksorth@yahoo.co.uk; 12Department of Orthopaedic Surgery, Graduate School of Medical Sciences, Kanazawa University, 13-1 Takara-machi, Kanazawa, Ishikawa 920-8641, Japan; tsuchi@med.kanazawa-u.ac.jp; 13Clinical Microbiology, University of Milan, 20100 Milano, Italy

**Keywords:** diagnosis, infection, joint prosthesis, PJI, definition, criteria, WAIOT

## Abstract

The definition of peri-prosthetic joint infection (PJI) has a strong impact on the diagnostic pathway and on treatment decisions. In the last decade, at least five different definitions of peri-prosthetic joint infection (PJI) have been proposed, each one with intrinsic limitations. In order to move a step forward, the World Association against Infection in Orthopedics and Trauma (W.A.I.O.T.) has studied a possible alternative solution, based on three parameters: 1. the relative ability of each diagnostic test or procedure to Rule OUT and/or to Rule IN a PJI; 2. the clinical presentation; 3. the distinction between pre/intra-operative findings and post-operative confirmation. According to the WAIOT definition, any positive Rule IN test (a test with a specificity > 90%) scores +1, while a negative Rule OUT test (a test with a sensitivity > 90%) scores −1. When a minimum of two Rule IN and two Rule OUT tests are performed in a given patient, the balance between positive and negative tests, interpreted in the light of the clinical presentation and of the post-operative findings, allows to identify five different conditions: High-Grade PJI (score ≥ 1), Low-Grade PJI (≥0), Biofilm-related implant malfunction, Contamination and No infection (all scoring < 0). The proposed definition leaves the physician free to choose among different tests with similar sensitivity or specificity, on the basis of medical, logistical and economic considerations, while novel tests or diagnostic procedures can be implemented in the definition at any time, provided that they meet the required sensitivity and/or specificity thresholds. Key procedures to confirm or to exclude the diagnosis of PJI remain post-operative histological and microbiological analysis; in this regard, given the biofilm-related nature of PJI, microbiological investigations should be conducted with proper sampling, closed transport systems, antibiofilm processing of tissue samples and explanted biomaterials, and prolonged cultures. The proposed WAIOT definition is the result of an international, multidisciplinary effort. Next step will be a large scale, multicenter clinical validation trial.

## 1. Introduction

“There is no single accepted set of diagnostic criteria for PJI. Various definitions have been proposed; however, none have been widely adopted. Furthermore, some of these definitions disagree with each other (…)”. Based on these considerations, the workgroup of Musculoskeletal Infection Society (MSIS) declared in 2011 “the intention (…) to have a “gold standard” definition for PJI” and proposed a set of major and minor criteria to define a peri-prosthetic joint infection (PJI) [1].

From a scientific point of view, the lack of a common definition of a clinical condition is a source of bias when comparing the results from different studies, weakening the power of systematic reviews and meta-analysis [2]. On the other hand, from a clinical perspective, defining a PJI plays a strategic role to decide the tests that need to be performed in a given patient; consequently, the definition of PJI is driving treatment and prognosis and has a strong impact on epidemiological, social, economic and medico-legal evaluations [3]. As a matter of fact, the lack of a shared PJI definition is limiting our ability to standardize the diagnostic pathway and the treatment choice.

However, in spite of the MSIS declaration and the recognized need for a univocal and shared definition of PJI, since 2011 at least four more formulas have been published, all from highly respected scientific institutions, including the Infectious Disease Society of America (IDSA) [4], two International Consensus Meetings [5,6,7] and, more recently, the European Bone and Joint Infection Society (EBJIS) [8].

To sum it up, in less than a decade at least five different definitions of PJI have been released, proposing a wide and heterogeneous set of criteria, scoring systems and reference values (Table 1). The difficulty of finding “a gold standard” relies on many factors. Peri-prosthetic infection has a wide range of clinical presentations, from the acute, high-grade to the subclinical low-grade one [8,9,10]; moreover, novel markers and diagnostic tools are continuously discovered [11,12,13], making any definition criteria quickly obsolete. Furthermore, in the lack of an accepted benchmark towards which the definition itself can be validated, the professional and scientific background of the panel of experts may bias their choice of the definition criteria. As a result, in spite of the several attempts, current definitions still appear limited in their ability to distinguish the different clinical presentations and even the most recent and complex scores may result as “Inconclusive” [6]. Moreover, imaging techniques, with a proven diagnostic accuracy, are not included [14], while laboratory tests with similar sensitivity or specificity cannot be used alternatively, thus restricting the choice for physicians that operate in a scarce resources environment and prompting the adoption of novel, often more expensive, biomarkers.

In this panorama, the WAIOT multidisciplinary Study Group on Bone and Joint Infection Definitions is proposing a novel approach to PJI definition, based on the relative ability of the available tests to confirm (rule in) or to exclude (rule out) a peri-prosthetic infection, applied to the various clinical presentations, with post-operative validation.

## 2. Pre- and Intra-Operative Rule IN and Rule OUT Test: Definition and Selection

The accuracy of any diagnostic test with continuous results, as the majority of those used for peri-prosthetic infection diagnosis are, depends on its cut-off value [15]. Choosing an appropriate cut-off value is then of paramount importance in using a test effectively.

Several criteria, mostly based on Receiver Operating Characteristic (ROC) curve analysis, have so far been proposed for choosing the most appropriate cut-off value [16]. Each point on a ROC curve corresponds to a cut-off value and is associated with a test True Positive rate (Sensitivity, Se) and a False Positive rate (1-Specificity, 1-Sp). Choosing the cut-off point for clinical application thus requires a compromise between desired Se and Sp. If there is no preference between Se and Sp, a commonly followed approach is to maximize both indices. Even with this procedure, however, any cut-off value that will be finally chosen will inevitably be associated with a better sensitivity or specificity. This fact should be taken into consideration when using a given test in the clinical setting.

For example, if we consider the serum C-Reactive Protein (C-RP) levels at a cut-off value of 10 mg/L, according to Ghanem and co-workers [17] this marker has a sensitivity of 91.1% (95% confidence interval [CI]: 85–95) and a specificity of only 76.6% (72–81) to diagnose a PJI. Hence, a negative C-RP test is quite reliable at excluding an infection, while a positive C-RP is not so accurate to confirm the disease. According to these findings, C-RP is then to be considered a good rule OUT test, but only a moderately reliable Rule IN test.

At the opposite, serum procalcitonin (PC), according to a recent meta-analysis [18], features only 58% (95% CI, 31–81) sensitivity but 95% specificity (95% CI, 63–100). On this basis, PC is to be considered a good Rule IN test, but not a useful Rule OUT examination.

For the purpose of the proposed WAIOT definition of PJI, all the tests and diagnostic procedures, that can be performed pre- or intra-operatively and that have been shown to have a sensitivity > 90%, are included as “Rule OUT” tests, while those with a specificity > 90% are considered “Rule IN” tests. If a test does show a sensitivity AND a specificity > 90%, it will be included both as a “Rule OUT” and “Rule IN” test. In order to select the Rule OUT and Rule IN tests, a literature search has been conducted by the WAIOT Study Group on the relevant search engines in order to assess the relative sensitivity and specificity of various tests and procedures commonly used to diagnose a PJI. In particular, we searched for all studies (retrospective, prospective, or randomized controlled trials) reporting on diagnosis of peri-prosthetic joint infection in MEDLINE, EMBASE, Web of Science, and Cochrane databases from inception up to December 2018. The search strategy used a combination of key words related to PJI with focus un diagnosis, serum and synovial markers, histology, imaging and microbiological techniques. Only papers in English were reviewed. We complemented the search by manually scanning reference lists of identified articles and review articles for relevant publications missed by the original search.

The selection has then been made through a progressively higher scientific evidence criteria, choosing first the cut-off and sensitivity/specificity values emerging from meta-analysis or systematic reviews and, if these were not available, the values reported from studies with the largest patients’ populations. Tests validated in cohorts of patients of less than 50 patients were excluded, while purulence, draining sinus and implant exposure were considered highly specific, even in the absence of a clinical trial to support this assumption. The results are reported in Table 2.

It is worth noting that the proposed definition of PJI allows one to add novel tests and markers of PJI at any time, provided that they meet the sensitivity/specificity threshold requirements mentioned above.

To define a PJI, any negative “Rule OUT” tests is scored −1, while any positive “Rule IN” test is scored +1. The score of a positive Rule OUT test or that of a negative Rule IN test is rated 0.

The rationale of this scoring modality relies on two basic observations. The first is that, by definition, two tests with similar sensitivity are equally effective at excluding a given condition. This observation grounds the basis to score all the negative Rule OUT tests the same (−1 point); a similar rationale does apply for scoring with the same value all the positive Rule IN tests.

The second observation is that, given the fact that the same threshold for sensitivity and specificity (>90%) was chosen to include a test as a Rule OUT or a Rule IN test, we may assume that the relative ability of a positive Rule IN test to confirm the infection and that of a negative Rule OUT test to exclude the infection are similar. This is why each negative Rule OUT tests score is scored −1 and each positive Rule IN test is rated exactly +1

## 3. Post-Operative, Confirmative Tests

Post-operative findings, including histology and microbiological analysis of tissue samples and of explanted biomaterials, may only become available, with current technologies, days after surgery. Even if these tests may not be used to drive surgical decision, their importance to confirm pre- and intra-operative findings is obvious from a clinical, epidemiological, research and medico-legal points of view.

Histology is important not only to confirm a possible PJI [30,31], but also to identify other reasons for implant failure, including the presence of metallosis or immunological reactions, as well as tumors [32,33,34]. With currently used thresholds, if correctly performed [29], histology is to be considered a highly specific investigation [35].

Similarly, microbiological analysis on tissue samples and on explanted biomaterials is mandatory in all cases of suspicion of infection. It not only provides a confirmation of the diagnosis, but also drives the antibiotic treatment.

Given the key confirmatory role played by microbiology in the present PJI definition, microbiological analysis should follow some basic rules, known to improve the results. The adherence to a correct sampling policy is strategic to reduce the occurrence of both false positive and false negative cultural examinations. In particular, concerning the latter, the presence of the biofilms on both the surface of the infected biomaterials and of contaminated tissues and the fact that bacteria may persist even in fluids as aggregates of biofilm-embedded microorganisms [36]

Concerning sampling, from 3 to 6 tissue samples should be taken, reducing the contamination risks with specific sampling tray and closed transport systems or blood culture bottles [37,38]; explanted biomaterials, including the prosthesis, modular parts and bone cement should also be sent for microbiological examination, using antibiofilm mechanical or chemical processing techniques [39,40,41]. Finally, cultures should be maintained for at least 14 days, in order to detect slow-growing microorganisms, while caution should be exerted when interpreting negative cultural examinations in patients that received antibiotics close or at the time of sampling [42,43,44,45].

Considering the importance of pathogen(s) identification and especially the need for identifying their antibiotic resistance profile, microbiology remains pivotal to confirm a PJI. Molecular assays, like for example MALDI-TOF mass spectrometry, Next Generation Sequencing (NGS) and other metagenomic procedures, that have been recently developed to provide a quick diagnosis, have not yet demonstrated their superiority compared to traditional cultural examination, in the clinical setting, but they may prove a useful tool in the next future [46,47].

## 4. WAIOT Definition of PJI

According to the proposed definition, a minimum of two “Rule OUT” and two “Rule IN” tests needs to be performed in any given patient, in order to define the presence of a peri-prosthetic joint infection.

Regarding the choice of the tests, there is not a fixed prescription and the clinician is left free to decide, in any given patient, on the basis of clinical, logistical and economic considerations. This recommendation takes into account the lack of scientific evidence showing a superiority between tests with similar sensitivity or specificity.

As for the number of the tests, provided that the minimum of two “Rule OUT” and two “Rule IN” is fulfilled, the physician is left free to perform more investigations, depending on the results and the clinical suspicion. The level of the clinical suspicion is directly related to the history of the patient, the clinical presentation and the presence of other conditions, that may explain the signs and symptoms in that patient. In fact, at variance with the other existing definitions of PJI, in our proposal the interpretation of the instrumental tests is performed also considering clinical aspects (cf. Table 3).

According to the proposed WAIOT definition, we can distinguish the following five conditions (cf. Table 3):

“**High-Grade PJI**” (HG-PJI)”. A clinical condition in which two or more signs or symptoms of local inflammation (pain, swelling, redness, warmth, *function laesa*) are present. This condition corresponds to what usually the clinicians call “acute” or “inflammatory” infection [11]; these patients are defined as affected by PJI if the balance between positive “Rule IN” and negative “Rule OUT” tests is ≥ 1. Generally, in these patients, only two Rule IN and two Rule OUT tests are needed, as often the main clinical problem here is not the diagnosis of infection, but the identification of the pathogen. The condition is considered post-operatively confirmed in case of positive histological and/or microbiological findings (at least one microbiological sample positive, microbiological analysis preferably performed with prolonged cultures and antibiofilm technique, with closed sampling and transport systems);“**Low-Grade PJI**” (LG-PJI). A condition in which, in the lack of acute local inflammatory signs and in the absence of an alternative explanation for the signs or symptoms, the patient complains for at least one of the followings: otherwise “unexplained” pain, swelling and/or reduced range of motion or functional impairment; often this condition is due to slowly growing microorganism(s) and it may show up several months or even years after surgery. If the balance between positive “Rule IN” and negative “Rule OUT” tests is ≥ 0 a Low-Grade PJI should be considered. In these patients often the clinician will prefer to perform additional tests, other than the minimum 2+2. The condition will be post-operatively proven in case of positive histology and/or if at least one microbiological sample is positive (microbiological analysis performed with prolonged cultures and antibiofilm technique, preferably using closed sampling and transport systems);“**Biofilm-related implant malfunction”** (BIM). A condition similar to LG-PJI, but where the balance between positive “Rule IN” tests and negative “Rule OUT” tests is < 0. In this case the diagnosis may only be confirmed post-operatively, based on positive histology and/or if at least one positive microbiological sample (microbiological analysis performed with prolonged cultures and antibiofilm technique, preferably using closed sampling and transport systems); often slow-growing microorganisms, like *Cutibacterium* (previously named *Propionibacterium*) *acnes*, should be expected. It should be noted that patients with BIM may only complain of moderate pain and/or reduced function, which may be not so severe to necessitate a reoperation, so a post-operative confirmation of the diagnosis may not be possible in all the cases.

On the average, the estimated incidence of high-grade infection lays in a range of 0.5% to 2.5% after clean surgery, while low-grade infections and biofilm-related malfunctions probably occur in approximately 5% to 10%. Beswick et al. [48] in a recent systematic review of prospective studies in patients undergoing total hip or knee replacement for osteoarthritis, found a proportion of people with long-term pain of unknown origin ranging from about 7% to 23% after total hip joint replacement and from 10% to 34% after knee replacement, while other studies have shown that “*between 4% and 13% of patients with preoperative diagnosis of aseptic loosening were infected*”, when retrieved implants were analyzed with genomic identification methods [49]. Furthermore, *C. acnes* has been identified in recent years as an occult causative agent of pain after shoulder prosthesis [50].

**Contamination** is said to occur in a patient with one or more condition, other than infection, able to motivate the signs or symptoms and the reason for reoperation (e.g., wear debris, recurrent dislocation or joint instability, fracture, malposition, neuropathic pain, etc.); in this patient the balance between positive “Rule IN” and negative “Rule OUT” tests is <0, while a microorganism is recovered from a single synovial fluid or intra-operative sample, but histology is negative.

**No infection** is here defined as a condition similar to that of “Contamination”, but with negative cultural examination.

Table 4 outlines some real case examples of patients defined as not infected or, respectively, affected by a Low- or a High-Grade-PJI.

## 5. Discussion

Through a multi-disciplinary and multi-national approach, in an effort to overcome some of the limitations of the existing score systems, we outlined here a novel approach to define peri-prosthetic joint infection.

The proposed WAIOT definition of PJI shows the following peculiarities.

First, to the best of our knowledge, this is the only to distinguish between clinical presentations, including high- and low-grade infections, biofilm-related implant malfunction, contamination and no infection with a unique scoring system. This distinction takes into account the most recent understanding of the biofilm-related nature of all the PJIs; [51] in fact, the ability of bacteria to persist in biofilms and even inside the host’s cells is among the main reasons for PJI clinical presentations characterized by no or few inflammatory signs and symptoms with conflicting diagnostic tests’ results [10,52,53].

A second peculiar feature is that the present definition is designed to maximize the use of each test, taking into account that the available diagnostic procedures may be more effective at confirming or at excluding a PJI.

A third specific feature is that all the tests that did demonstrate similar sensitivity and specificity are put on the same level, leaving the choice to the clinician, on the basis of medical, logistical and economic considerations.

As a matter of fact, current diagnostic pathways are extremely variable and differentiated even among the centers of excellence around the world [54]. This not only makes the diagnosis less predictable and homogeneous, but it may also lead to inappropriate use of the available diagnostic tests, that may be under- or overused. According to a recent trial, the average number of tests performed to confirm a PJI is 5.8 [55]. The present definition reduces the number of tests needed to define a PJI to a minimum of two Rule IN and two Rule OUT tests.

Finally, the proposed definition differentiates between a pre-/intra-operative tests and post-operative histological and microbiological confirmatory findings.

As a practical consequence, the diagnostic pathways can be differentiated, according to the clinical presentation, allowing some flexibility to the clinician and reserving the most expensive and sophisticated tests only to those cases in which the diagnosis is particularly challenging. Furthermore, this approach allows to include novel tests or procedures, provided that they meet the minimum sensitivity and specificity, without the need for changing the definition structure.

The proposed definition has its own limitations. First, the selection of the tests is based on the existing literature. The search itself may contain mistakes and be not completely exhaustive; on the other hand, the cut-off value and the diagnostic accuracy of each test has been investigated according to different benchmarks and methodologies, causing an intrinsic bias in ours and in any definition that relies on previous studies. In this regard, it should also be noted that some commonly accepted tests are only based on semi-quantitative results (e.g., leukocyte esterase strips) or may be subject to interpretation bias (e.g., nuclear medicine imaging) [56].

The second limitation is the lack of clinical validation. As far as we know, this limit is shared by all the current definitions [1,4,5,8], except maybe the latest one [7]. In this regard it is worth noting that the definition criteria and the score, voted at the last Consensus Meeting in Philadelphia in 2018 [7], do not exactly match the criteria used in the clinical validation study [6]. In particular, synovial fluid markers were scored differently and also the definition of infection according to the “minor criteria” was different. Furthermore, the agreement on this definition has been “weak” (68%) [8]. In our opinion, this lack of consensus was due to several reasons, including, among others, the lack of a reliable benchmark in the validation study. In fact, in their retrospective analysis, the authors report that “*patients were classified as having a PJI if they met major diagnostic criteria of MSIS*”; [6] this means that all those patients that did not present with a draining sinus or with two positive cultural examinations were considered as not infected, thus excluding, “by definition” all the patients with a “low-grade infection” and with difficult-to-diagnose pathogens.

Aware of the challenges posed by clinical validations, our future target is to first retrospectively apply the proposed WAIOT definition to a large population of patients, comparing the results with other existing definitions; as a second step a prospective multinational trial aimed at comparing pre-operative and intra-operative application of the definition with post-operative data is planned.

In conclusion, a simplified definition of PJI, based on the most commonly observed clinical presentations and on the relative sensitivity and specificity of the available pre/intra-operative tests, with post-operative confirmation, may be helpful for research applications and for the clinicians willing to differentiate patients with PJI along with a cost-effective use of the available resources.

From a clinical perspective, this paper highlights the limits and the heterogeneity of the current definitions of PJI, an observation that needs to be considered in the diagnostic process. Furthermore, the proposed new definition allows for the first time to distinguish between high- and low-grade peri-prosthetic joint infections. This may be helpful in order to raise in the clinicians the legitimate suspicion that at least some cases of pain of “unknown” origin after joint replacement may, in fact, be difficult-to-diagnose infections; this, in turn, will hopefully trigger the appropriate type and number of tests that are needed to come to a correct definition and will consequently will reduce misdiagnosis and misinterpretation of the data.

## Figures and Tables

**Table 1 jcm-08-00650-t001:** Comparison of the diagnostic criteria, adopted in five peri-prosthetic joint infection (PJI) definitions, published from 2011 to 2018.

Definition Source	MSIS 2011 [1]	IDSA 2013 [4]	ICM 2013 [5]	ICM 2018 [7]	Proposed EBJIS 2018 [8]
**Scoring system**	1 of the 2 Major Criteria OR ≥4 of 6 Minor Criteria *	≥1 Positive Criteria *	1 of the 2 Major Criteria OR ≥3 of 5 Minor Criteria *	1 of the 2 Major Criteria OR Minor criteria scoring ≥6 Infected 3–5 Possibly infected (“Consider further molecular diagnostics such as next-generation sequencing”) <3 Not infected *	≥1 Positive Criteria
	* “PJI may be present if fewer than four of these criteria are met”	* “The presence of PJI is possible even if the above criteria are not met (…)”	* “PJI may be present without meeting these criteria, (…).”	* “Proceed with caution in: adverse local tissue reaction, crystal deposition disease, slow growing organisms”	
**Criteria**	Major: Sinus tract communicating with the prosthesis; A pathogen is isolated by culture from at least two separate tissue or fluid samples obtained from the affected prosthetic joint Minor: (a) Elevated ESR (>30 mm/hr) and CRP (>10 mg/L) concentration (b) Elevated synovial leukocyte count (c) Elevated PMN% (d) Purulence in the affected joint (e) Isolation of a microorganism in one culture of periprosthetic tissue or fluid (f) Greater than five neutrophils per high-power field in five high-power fields observed from histologic analysis of periprosthetic tissue at x400 magnification	Sinus tract communicating with the prosthesis Purulence without other etiology surrounding the prosthesis Acute inflammation seen on histopathological examination of the periprosthetic tissue ≥ 2 intraoperative cultures or combination of preoperative aspiration and intraoperative cultures yielding an indistinguishable organism [the growth of a virulent microorganism (e.g., Staphylococcus aureus) in a single specimen of a tissue biopsy or synovial fluid is also considered as indicative of a PJI]	Major A sinus tract communicating with the joint Two positive periprosthetic cultures with phenotypically identical organisms, Minor: (a) Elevated ESR (>30 mm/hr) and CRP (>100 mg/L for acute infections; >10 mg/L for chronic infections) (b) Elevated synovial fluid WBC count (>10,000 cells/mL for acute infections; >3,000 cells/mL for chronic infections) or ++ change on leukocyte esterase test strip (c) Elevated PMN% (>90% for acute infections; >80% for chronic infections) (d) Positive histological analysis of periprosthetic tissue (> 5 neutrophils per high-power field in five high-power fields observed on periprosthetic tissue at x400 magnification) (e) A single positive culture	Major: Sinus tract with evidence of communication to the joint or visualization of the prosthesis Two positive growths of the same organism using standard culture methods Minor: (a) Elevated CRP (>100 mg/L for acute infections; >10 mg/L for chronic infections) or D-Dimer (unknown threshold for acute infection; >860 ug/L for chronic infection) (score 2) (b) Elevated ESR (no role for acute infections; >30 mm/hr for chronic infections) (score 1) (c) Elevated synovial WBC count (>10,000 cells/mL for acute infections; >3,000 cells/mL for chronic infections) OR Leukocyte Esterase (++ for acute and chronic infections) OR Positive alpha-defensin (score 3) (d) Elevated synovial PMN% (>90% for acute infections; >70% for chronic infections) (score 2) (e) Single positive culture (score 2) (f) Positive histology (score 3) (g) Positive intraoperative purulence (score 3)	Purulence around the prosthesis or sinus tract Increase synovial fluid leukocyte count (>2,000 cells/mL or >70 % granulocytes) Positive histopathology Confirmatory microbial growth in synovial fluid, periprosthetic tissue, or sonication culture (“Confirmatory microbial growth in periprosthetic tissue: if positive in ≥1 specimen in highly virulent organisms or ≥ 2 in low virulent pathogens; sonication culture considered positive if >50 colony-forming units/mL of sonication fluid.”)

**Table 2 jcm-08-00650-t002:** Pre- and intra-operative tests, classified according to their sensitivity and specificity and hence their ability to exclude (“Rule OUT”) or to confirm (“Rule IN”) a PJI. In parenthesis, the reference cut-off value considered here.

**Rule OUT Tests (Sensitivity > 90%)** **EACH NEGATIVE TEST Scores −1** **(Positive Rule OUT Tests Score 0)**	
Serum	ESR (>30 mm/h) [17] CRP (>10 mg/L) [17]
Synovial fluid	WBC (>1,500/μL) [19] LE (++) [20,21] Alpha-Defensin immunoassay (>5.2 mg/L) [22,23]
Imaging	Tc99 bone scan [24]
**Rule IN Tests (Specificity > 90%)** **EACH POSITIVE TEST Scores +1** **(Negative Rule IN Tests Score 0)**	
Clinical examination	Purulence or draining sinus or exposed joint prosthesis
Serum	IL-6 (>10 pg/mL) [18] PC (>0.5 ng/mL) [18] D-Dimer (>850 ng/mL) [25]
Synovial fluid	Cultural examination [26] WBC (>3,000/mL) [19,27] LE (++) [20,21] Alpha-Defensin immunoassay (>5.2 mg/L) [22,23] or lateral flow test [28]
Imaging	Combined leukocyte and bone marrow scintigraphy [24]
Histology	Frozen section (5 neutrophils in at least 3 HPFs) [29]

Abbreviations: ESR: erythrocyte sedimentation rate; CRP: C-Reactive Protein; IL-6: Interleukin-6; WBC: White blood cell count; PC: Procalcitonin; LE: Leukocyte esterase strip (++); HPFs: high power fields (×400).

**Table 3 jcm-08-00650-t003:** WAIOT proposed definition of peri-prosthetic joint infection (PJI).

	No Infection	Contamination	BIM	LG-PJI	HG-PJI
Clinical presentation	One or more condition(s), other than infection, can cause the symptoms or the reason for reoperation (e.g., wear debris, metallosis, recurrent dislocation or joint instability, fracture, malposition, neuropathic pain)		One or more of the followings: otherwise “unexplained” pain, swelling, stiffness		Two or more of the followings: pain, swelling, redness, warmth, *functio laesa*
# of Positive Rule IN minus # of Negative Rule OUT tests	<0	<0	<0	≥0	≥1
Post-operatively confirmed if	Negative cultural examination	One pre- or intra-operative positive culture, with negative histology	Positive cultural examination (preferably with antibiofilm techniques) and/or positive histology		

Abbreviations: WAIOT: World Association against Infection in Orthopedics and Trauma; BIM: Biofilm-related Implant malfunction; LG-PJI: Low-Grade Peri-Prosthetic Joint Infection; HG-PJI: High-Grade Peri-Prosthetic Joint Infection.

**Table 4 jcm-08-00650-t004:** Real case examples of diagnostic pathways, that can be used to confirm or exclude PJI, according to the WAIOT proposed definition (n.p.: not performed).

		**Case 1.** **No Infection**		**Case 2.** **Low-Grade PJI**		**Case 3.** **High-Grade PJI**	
		**Clinical presentation**					
		Male, 48 years, recurrent hip arthroplasty dislocation, undergoing partial revision 8 months after joint replacement.		Female, 72 years, continuous kneepain 12 months after joint replacement.		Male, 57 years, local redness, pain, and swelling, 13 months after total hip replacement	
		**Pre/Intra-operative Tests**					
**Rule OUT Tests** **(If Negative Score −1)**		**Result**	**Score**	**Result**	**Score**	**Result**	**Score**
Serum	ESR (>30 mm/h)	Positive	0	Positive	0	Positive	0
	CRP (>10 mg/L)	**Negative**	**−1**	**Negative**	**−1**	Positive	0
Synovial fluid	WBC (>1,500/μL)	**Negative**	**−1**	**Negative**	**−1**	Positive	0
	LE (++)	**Negative**	**−1**	Positive	0	Positive	0
	Alpha-Defensin immunoassay (>5.2 mg/L)	n.p.		n.p.		n.p.	
Imaging	Tc99 bone scan	n.p.		Positive	0	n.p.	
**Rule IN Tests** **(If Positive Score +1)**		**Result**	**Score**	**Result**	**Score**	**Result**	**Score**
Serum	IL-6 (>10 pg/mL)	n.p.		Negative	0	n.p.	
	PC (>0.5 ng/mL)	n.p.		n.p.		n.p.	
	D-Dimer (>850 ng/mL)	n.p.		n.p.		n.p.	
Clinical examination	Purulence or draining sinus or exposed joint prosthesis	Negative	0	Negative	0	Negative	0
Synovial fluid	Cultural examination	Negative	0	Negative	0	Negative	0
	WBC (>3,000/mL)	Negative	0	Negative	0	**Positive**	**+1**
	LE (++)	Negative	0	**Positive**	**+1**	**Positive**	**+1**
	Alpha-Defensin immunoassay (>5.2 mg/L) or lateral flow test	n.p.		n.p.		n.p.	
Imaging	Combined leukocyte and bone marrow scintigraphy	n.p.		**Positive**	**+1**	n.p.	0
Histology	Frozen section (5 neutrophils in at least 3 HPFs)	n.p.		Negative	0	n.p.	0
**TOTAL SCORE**		**−3**		**0**		**+ 2**	
**Pre-Intra Operative Interpretation**		**No Infection**		**Low-Grade PJI**		**High-Grade PJI**	
		**Post-Operative Tests**					
Histology		n.p.		Negative		**Positive**	
Microbiology		**Negative**		**Positive**		**Positive**	
**Post-Operative Interpretation**		**No Infection Confirmed**		**LG PJI Confirmed**		**HG PJI Confirmed**	

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
