# Peer review of "The W.A.I.O.T. Definition of High-Grade and Low-Grade Peri-Prosthetic Joint Infection"

_jcm, 2019, doi:10.3390/jcm8050650_

Reviewer 1 Report

To the authors: 

Thank you for this concept paper. My thoughts are below.

Introduction: Well, written and concise. My thoughts are they they should mention something about how a "gold standard" in definition is difficult when there is still no gold standard treatment or test, and when final definition is often only decided upon after re-operation. 

Pre- and Intra-operative rule IN and rule OUT test: definition and selection- This is an important section. It is again, very well-written and very clear. 

Post-operative, confirmative tests:  This section is important, but I would ask if the authors have any comments on how often cultures are culture-negative, and WHY this might be the case. Consider referring to the research done on bacterial aggregation, biofloat formation, and quiescence, particularly of certain more common species of bacteria  in synovial fluids. These aggregates are resistant to culture and resistant to antibiotic treatment.  ( Dastgheyb et al. 2014)

Also consider listing examples of molecular assess that the authors refer to has having " recently been developed"

WAIOT definition of PJI: please touch on if the need to perform a sufficient number of tests to obtain a score will result in increased spending. What is the average number of tests the current clinicians runs to diagnose a PJI? Are they  "good" tests? what are the sensitivity and specificity of the top 10 tests that are run to diagnose PJI?

Discussion:  It might help if you would make clinical recommendations, although I appreciation that this is purely a definition discussion, but guidelines maybe more appropriate as this is the Journal of Clinical Medicine. 

Where you say "do not match the criteria used in the clinical validation study"- how so?

Ultimately, how important is this clinically?

Citations 8 and 55 are the SAME.

Take out the last part of the 3rd from last paragraph in the discussion.  basically from where you say " on the other hand......... main challenges that bias them" none of this is necessary.

Where you say " design a novel type of prospective multinational trial"- please comment on how you believe it could be designed- potentially propose a figure. I don't think that this paper necessarily needs that to be published, although it would be a nice supplemental figure. 

Table 1 is excellent. 

Table 2- This could use some work. Title is odd. "or intra-operative tests"? not sure what that means.  LE (++) needs to be redefined by count, Imaging Tc99 scan needs to report a reading value. I'm not sure if D-Dimer is actually >90% specific. In what population?culture examination of synovial fluid- be more specific. culture result? gram stain? 

Table 3- fine

Table 4- This is the most important teaching point. Very appreciated.

Author Response

Reviewer 1

To the authors: 

Thank you for this concept paper. My thoughts are below.

Introduction: Well, written and concise. My thoughts are they they should mention something about how a "gold standard" in definition is difficult when there is still no gold standard treatment or test, and when final definition is often only decided upon after re-operation. 

Thank you for this comment. The following sentence has now been added to the text:

As a matter of fact, the lack of a shared PJI definition is limiting our ability to standardize the diagnostic pathways and the treatment choice.”

Pre- and Intra-operative rule IN and rule OUT test: definition and selection- This is an important section. It is again, very well-written and very clear. 

Thank you.

Post-operative, confirmative tests:  This section is important, but I would ask if the authors have any comments on how often cultures are culture-negative, and WHY this might be the case. Consider referring to the research done on bacterial aggregation, biofloat formation, and quiescence, particularly of certain more common species of bacteria  in synovial fluids. These aggregates are resistant to culture and resistant to antibiotic treatment.  ( Dastgheyb et al. 2014)

Thank you for this very appropriate comment. The citation is now added with the following sentence::

The adherence to a correct sampling policy is strategic to reduce the occurrence of both false positive and false negative cultural examinations. In particular, concerning the latter, the presence of the biofilms on both the surface of the infected biomaterials and of contaminated tissues and the fact that bacteria may persist even in fluids as aggregates of biofilm-embedded microorganisms [37].”

Also consider listing examples of molecular assess that the authors refer to has having " recently been developed"

The following sentence is now added to the text:

like for example MALDI-TOF mass spectrometry, Next Generation Sequencing (NGS) and other metagenomic procedures”

WAIOT definition of PJI: please touch on if the need to perform a sufficient number of tests to obtain a score will result in increased spending. What is the average number of tests the current clinicians runs to diagnose a PJI? Are they  "good" tests? what are the sensitivity and specificity of the top 10 tests that are run to diagnose PJI?

Thank you for giving us the chance to clarify this point. The following sentence has been added to the Discussion section::

As a matter of fact, current diagnostic pathways are extremely variable and differentiated even among the centers of excellence around the world [cf. Drago L, Lidgren L, Bottinelli E, et al. Mapping of Microbiological Procedures by the Members of the International Society of Orthopaedic Centers (ISOC) for Diagnosis of Periprosthetic Infections. J Clin Microbiol. 2016;54(5):1402–1403. doi:10.1128/JCM.00155-16]. This not only makes the diagnosis less predictable and homogeneous, but it may also lead to inappropriate use of the available diagnostic tests, that may be under- or overused. According to a recent trial, the average number of tests performed to confirm a PJI is 5.8 [cf. Gallazzi E, Drago L, Baldini A, et al. "Combined Diagnostic Tool" APPlication to a Retrospective Series of Patients Undergoing Total Joint Revision Surgery. J Bone Jt Infect. 2017;2(2):107–113. Published 2017 Feb 4. doi:10.7150/jbji.18308]. The present definition reduces the number of tests needed to define a PJI to a minimum of two Rule IN and two Rule OUT tests.”

Discussion:  It might help if you would make clinical recommendations, although I appreciation that this is purely a definition discussion, but guidelines maybe more appropriate as this is the Journal of Clinical Medicine. 

Thank you for this comment. Since this is a concept paper we are not going to propose the new definition as a guideline, however, the specific features and advantages of adopting it are listed in the Discussion section. We have not also added the following new paragraph, that summarizes what is the most interesting innovation brought forward by the novel definition:

From a clinical perspective, this paper highlights the limits and the heterogeneity of the current definitions of PJI, an observation that needs to be considered in the diagnostic process. Furthermore, the proposed new definition allows for the first time to distinguish between high- and low-grade peri-prosthetic joint infections. This may be helpful in order to raise in the clinicians the legitimate suspicion that at least some cases of pain of “unknown” origin after joint replacement may in fact be difficult-to-diagnose infections; this in turn will hopefully trigger the appropriate type and number of tests that are needed to come to a correct definition and will consequently will reduce misdiagnosis and misinterpretation of the data.

Where you say "do not match the criteria used in the clinical validation study"- how so?

The criteria voted at the Consensus do not match those used in the validation clinical study published by Parvizi and co-workers.

In particular, in the clinical validation study a positive Alpha-defensin alone was scored 3 points and an “Elevated Synovial WBC or Leukocyte Esterase” were also rate as 3. In this way a positive Alpha-defensin would sum up 3 with a positive Synovial WBC or Leukocyte Esterase.

In the definition voted at the Consensus, a positive Alpha-defensin is included as one of the three tests (“Elevated Synovial WBC or Leukocyte Esterase or Positive Alpha-defensin”), so that if at least one of the three is positive, the score is 3.

Furthermore, according to the minor criteria of the validation study the patient is “possibly infected” with a pre-operative score of 2-5 and not infected with a score < 1, while in the Consensus definition the patient is “possibly infected” with a score of 3-5 and not infected with a score < 3.

The following sentence has now been included in the Discussion:

“In particular, synovial fluid markers were scored differently and also the definition of infection according to the “minor criteria” was different.”

Ultimately, how important is this clinically?

Please refer to the new paragraph mentioned above:

From a clinical perspective, this paper highlights the limits and the heterogeneity of the current definitions of PJI, an observation that needs to be considered in the diagnostic process. Furthermore, the proposed new definition allows for the first time to distinguish between high- and low-grade peri-prosthetic joint infections. This may be helpful in order to raise in the clinicians the legitimate suspicion that at least some cases of pain of “unknown” origin after joint replacement may in fact be difficult-to-diagnose infections; this in turn will hopefully trigger the appropriate type and number of tests that are needed to come to a correct definition and will consequently will reduce misdiagnosis and misinterpretation of the data.

Citations 8 and 55 are the SAME.

Corrected, thank you.

Take out the last part of the 3rd from last paragraph in the discussion.  basically from where you say " on the other hand......... main challenges that bias them" none of this is necessary.

Now removed, thank you.

Where you say " design a novel type of prospective multinational trial"- please comment on how you believe it could be designed- potentially propose a figure. I don't think that this paper necessarily needs that to be published, although it would be a nice supplemental figure. 

Actually we are now first going to validate the definition with a retrospective international study and then we are going to start the prospective trial. This is now explained in the sentence rephrased as below:

Aware of the challenges posed by clinical validations, our future target is to first retrospectively  apply the proposed WAIOT definition to a large population of patients, comparing the results with other existing definitions; as a second step a prospective multinational trial aimed at comparing pre-operative and intra-operative application of the definition with post-operative data is planned”.

Table 1 is excellent. 

Thank you.

Table 2- This could use some work. Title is odd. "or intra-operative tests"? not sure what that means.  

Thank you for the comment. This is a typing error. Now corrected to: “Pre- and intra-operative tests”

LE (++) needs to be redefined by count,

In all available papers LE is only assessed using the semiquantitative tests currently used for urine analysis; there is not quantitative tests validated or studied, to the best of our knowledge.

Imaging Tc99 scan needs to report a reading value. 

Once again, as far as we found this is not univocally mentioned in the available papers and, in particular, in the systematic analysis. This bias is now made clear in the Discussion section:

In this regard it should also be noted that some commonly accepted tests are only based on semi-quantitative results (e.g. leukocyte esterase strips) or may be subject to interpretation bias (e.g. nuclear medicine imaging) [Busby LP, Courtier JL, Glastonbury CM. Bias in Radiology: The How and Why of Misses and Misinterpretations. Radiographics. 2018 Jan-Feb;38(1):236-247. doi:10.1148/rg.2018170107. Epub 2017 Dec 1.].”

I'm not sure if D-Dimer is actually >90% specific. In what population?

This is the result from the Parvizi group on a cohort of 245 patients. They set the cut-off at a very high level (850 ng/mL), which probably allowed them to obtain such a high specificity. The paper is cited in the text and can be found at https://www.ncbi.nlm.nih.gov/pubmed/28872523.

culture examination of synovial fluid- be more specific. culture result? gram stain? 

Now corrected to “Cultural examination”, thank you. Gram stain is not used.

Table 3- fine

Thank you.

Table 4- This is the most important teaching point. Very appreciated. 

Thank you.

Reviewer 2 Report

Good overall definition for high grade and low grade periprosthetic joint infection grading.

It has developed a unique mechanism of rating the  severity of prosthetic joint infections.

The main question is addressed rating prosthetic joint infection on a scale.

This paper is interesting

Original topic not been investigated to this extend before

The paper is well written

Conclusions are consistent

The main question posed

I happy with journal article as outlines.

Author Response

Reviewer 2

Good overall definition for high grade and low grade periprosthetic joint infection grading.

It has developed a unique mechanism of rating the  severity of prosthetic joint infections.

The main question is addressed rating prosthetic joint infection on a scale.

This paper is interesting

Original topic not been investigated to this extend before

The paper is well written

Conclusions are consistent

The main question posed

I happy with journal article as outlines.

We thank Reviewer 2 for taking the time to read and for the encouraging comments..

Reviewer 3 Report

        In the present study, a group of authors belonging to the World Association Against Infection and Orthopedics and Trauma (WAIOT) proposes a new definition of prosthetic joint infection (PJI). This definition has characteristics of great interest such as: inclusion of clinical presentation data (not only “the fistula”), performing a presumptive diagnosis based on pre- and intra-operative tests, and a definitive posterior diagnosis (based on post-operative tests), as well as the attribution of negative or positive points (that is, against or in favor to a PJI diagnosis) to the presence or absence of some results of the diagnostic tests.

 COMMENTS

The attribution of positive or negative points to the results of different diagnostic tests is of great interest. However, the justification for the use of this methodology is missing. Is it an arbitrary decision of the authors or is it based on some recommendation / methodological consideration on the interpretation of diagnostic tests? It would be necessary to explain how the authors arrive at the current scoring proposal.

Regarding the literature research on PJI diagnostic tests, it would be recommendable to specify more how the research was done: the terms used, databases, languages...  

Early prosthetic radiological loosening could be also included among the clinical presentation data of PJI suspicion, since it is a sign that has been shown to be of diagnostic utility

The potential diagnostic role of PET-TC could be considered and analyzed.

Previous use of antibiotics can prevent the isolation of the causative microorganism of PJI, as it is well known. Despite this, however, antibiotics are sometimes used for different reasons (suspicion of another infection, initiation by other doctors ...); this is particularly problematic in the case of acute infections, where it may be difficult to suspend them for a sufficient period of time. No definition has taken this fact into account and its potential impact on the culture results. A comment in this regard would be advisable (caution when interpreting negative cultures if the patient has received antibiotics recently).

Distinction between High-Grade PJI (HG-PJI) and Low-Grade PJI (LG-PJI) might have clinical sense, making reference to different forms of PJI presentation (acute and chronic, respectively); additionally, acute and chronic infections are usually produced by different groups of microorganisms, which are usually categorized as high- or low-virulent organisms. However, this distinction presents some doubts:

First, it is not entirely clear, what are the practical consequences of arriving at defining a HG-PJI versus a LG-PJI.

The distinction between high-and low-grade infection is somewhat arbitrary and the definitions of HG-PJI and LG-PJI could overlap: in fact, the scores corresponding to the “rule IN and rule Out” do overlap  (≥ 0 and  ≥ 1) and a clinical picture of pain and swelling could be included in both definitions.

Finally, the definitions of HG-PJI and LG-PJI seem similar to those of acute and chronic infection, which refer to different forms of clinical presentation. However, the concepts of “High-Grade and Low-Grade infection” do not make a clear sense from the infectious point of view, since it is not clear whether it is referred to infections caused by more or less virulent microorganisms or a higher or lower bacterial load or both (or to a different thing) 

Both definitions could be contained within into a single category of PJI, which would include the different forms of clinical presentation, both chronic and acute.

The definition of "Biofilm-related Implant Malfunction (BIM)" is ambiguous. It seems to refer to a situation in which, although the clinical presentation includes the possible diagnosis of a chronic infection, the pre- and intra-operative are not conclusive, and the diagnosis of infection is achieved postoperatively. Therefore, this entity is finally a PJI, and not another different entity.

Probably the inclusion of 3 concepts: non-infection, contamination and PJI would be enough. The “PJI category” would include the possible clinical presentations mentioned (chronic and acute); the pre-and intraoperative results could be less than 0, not suggesting in principle an infection, equal than 0, suggesting a possible infection, or more than 0, indicating a high suspicion of infection.

Do the cases included in the Table 4 correspond to real patients? It would be advisable to indicate it. Is surprising that the Case 2 has a synovial fluid WBC negative, but the leukocyte esterase is positive; given the correlation between the two markers, it should be infrequent.

 FORMAL ASPECTS

In the Introduction, line 81, it is mentioned the potential “conflict of interests” of the experts involved in the realization of previous PJI, definitions. I guess that refers to the inclusion of the alpha-defensin as a criterion for the ICM 2018 PJI definiton; beyond this criterion, I see no clear conflicts of interest in previous definitions. Such a general expression about “conflict of interests” raises doubts about all the previous definitions that do not seem to conform to reality. Therefore, this mention should be reconsidered or narrow down.

In the Introduction, Lines 85-87 it is said "while laboratory tests with similar sensitivity or specificity cannot be used alternatively, thus restricted the choice for physicians that operate in scarce resources environment and prompting the adoption of novel, often more expensive, biomarkers". Again, it seems to refer to the ICM 2018 definition. However, being fair, this definition includes as similar alternatives "elevations of synovial WBC count, leukocite esterase and alpha-Defensin"; of these 3 tests, only the last one is an expensive biomarker and is not included as essential. The rest of the PJI definitions don't includes expensive tests. So, these sentences should be rewritten.

It is not clear the relationship between the mentioned in lines 63-65 and the reference number 3.

Table 1: It would be advisable to improve and order the presentation of this table:

Similar criteria should be presented in the same row and expressed in a similar way to facilitate the comparison between definitions. The MSIS 2011 and ICM 2013 definitions practically only differ in the criterion of the periprosthetic purulence and yet they seem very different definitions at first sight

Whenever the definition includes cut-off of quantitative variables, these should be shown (in table or table-foot)

The ICM 2018 definition gives different cut-offs in the case of acute and chronic infections, which should be mentioned

The ICM 2018 definition do not refers to "criteria" but points or score

The IDSA definition includes the possibility of a single culture with a virulent microorganism as a diagnostic criterion, which should be also mentioned

Table 2: the title must correspond to "Pre-" or Intraoperative Tests (it seems that the "pre-" was left out by mistake.

Table 3: The acronyms used should be defined at the table footnote

Line 130: The first time a microorganism is mentioned, the genus must be written without shortening (here, it would not be right "C. acnes”); on the other hand, currently Propionibacterium acnes is named Cutibacterium acnes. So, here it is recommended: Cutibacterium (previously named Propionibacterium) acnes.

Line 140: Here, however, it would be appropriate: "C. acnes".

In the Discussion is properly mentioned the difficulty of validating the PJI definitions for various reasons. It is criticized, in this sense, the validation carried out of the 2018 ICM definition. It is certainly a definition and a validation that presents problematic aspects. However, it is not clear how the authors of the current definition seek to circumvent the difficulties associated with the validation of their definition and whether this validation will be better than the previously mentioned. Additionally, it should be avoided to make literal quotes of what other authors mention and reference it with an indirect style. These aspects should be taken into account in the discussion.

Author Response

Reviewer 3

In the present study, a group of authors belonging to the World Association Against Infection and Orthopedics and Trauma (WAIOT) proposes a new definition of prosthetic joint infection (PJI). This definition has characteristics of great interest such as: inclusion of clinical presentation data (not only “the fistula”), performing a presumptive diagnosis based on pre- and intra-operative tests, and a definitive posterior diagnosis (based on post-operative tests), as well as the attribution of negative or positive points (that is, against or in favor to a PJI diagnosis) to the presence or absence of some results of the diagnostic tests.

COMMENTS

The attribution of positive or negative points to the results of different diagnostic tests is of great interest. However, the justification for the use of this methodology is missing. Is it an arbitrary decision of the authors or is it based on some recommendation / methodological consideration on the interpretation of diagnostic tests? It would be necessary to explain how the authors arrive at the current scoring proposal.

This is an appropriate question and we thank the Review 3, that gives us the opportunity to better clarify the rationale of our approach. This is now reported at the end of the section “Pre- and Intra-oerative Rule IN and Rule OUT test: definition and selection”:

The rationale of this scoring modality relies on two basic observations. The first is that, by definition, two tests with similar sensitivity are equally effective at excluding a given condition. This observation grounds the basis to score all the negative Rule OUT tests the same (- 1 point); a similar rationale does apply for scoring with the same value all the positive Rule OUT tests. 

The second observation is that, given the fact that the same threshold for sensitivity and specificity (> 90%) was chosen to include an exam as a Rule OUT or Rule IN test, we may assume that the relative ability of a positive Rule IN test to confirm the infection and that of a negative Rule OUT test to exclude the infection are similar. This is why each negative Rule OUT tests score is scored – 1 and each positive Rule IN test is rated exactly + 1.”

Regarding the literature research on PJI diagnostic tests, it would be recommendable to specify more how the research was done: the terms used, databases, languages... 

Thank you for this important point. We have now added the following sentence:

In particular, we searched for all studies (retrospective, prospective, or randomized controlled trials) reporting on diagnosis of peri-prosthetic joint infection in MEDLINE, EMBASE, Web of Science, and Cochrane databases from inception up to December 2019. The search strategy used a combination of key words related to PJI with focus un diagnosis, serum and synovial markers, histology, imaging and microbiological techniques. Only papers in English were reviewed. We complemented the search by manually scanning reference lists of identified articles and review articles for relevant publications missed by the original search.”

Early prosthetic radiological loosening could be also included among the clinical presentation data of PJI suspicion, since it is a sign that has been shown to be of diagnostic utility

This is a good observation. However, we decided only to include as “clinical presentation” the clinical signs and symptoms. Introducing “early radiological loosening2 would require further definition of what is intended as “early” and all the differential radiological diagnosis that may cause osteolysis around the implant.

The potential diagnostic role of PET-TC could be considered and analyzed.

Thank you for this good point. However, we really looked at studies on this specific technology, but we could not find any relevant trial focused on PJI diagnosis that met our inclusion criteria. This is why PET-TC is not mentioned; should there be a reference in this regard that we missed we would be very grateful to know and include.

Previous use of antibiotics can prevent the isolation of the causative microorganism of PJI, as it is well known. Despite this, however, antibiotics are sometimes used for different reasons (suspicion of another infection, initiation by other doctors ...); this is particularly problematic in the case of acute infections, where it may be difficult to suspend them for a sufficient period of time. No definition has taken this fact into account and its potential impact on the culture results. A comment in this regard would be advisable (caution when interpreting negative cultures if the patient has received antibiotics recently).

Correct. Now made clear with the following sentence:

Finally, cultures should be maintained for at least 14 days, in order to detect slow-growing microorganisms, while caution should be exerted when interpreting negative cultural examinations in patients that received antibiotics close or at the time of sampling”

Distinction between High-Grade PJI (HG-PJI) and Low-Grade PJI (LG-PJI) might have clinical sense, making reference to different forms of PJI presentation (acute and chronic, respectively); additionally, acute and chronic infections are usually produced by different groups of microorganisms, which are usually categorized as high- or low-virulent organisms. However, this distinction presents some doubts:

First, it is not entirely clear, what are the practical consequences of arriving at defining a HG-PJI versus a LG-PJI.

The distinction between high-and low-grade infection is somewhat arbitrary and the definitions of HG-PJI and LG-PJI could overlap: in fact, the scores corresponding to the “rule IN and rule Out” do overlap  (≥ 0 and  ≥ 1) and a clinical picture of pain and swelling could be included in both definitions.

Finally, the definitions of HG-PJI and LG-PJI seem similar to those of acute and chronic infection, which refer to different forms of clinical presentation. However, the concepts of “High-Grade and Low-Grade infection” do not make a clear sense from the infectious point of view, since it is not clear whether it is referred to infections caused by more or less virulent microorganisms or a higher or lower bacterial load or both (or to a different thing)

Both definitions could be contained within into a single category of PJI, which would include the different forms of clinical presentation, both chronic and acute.

This are all interesting comments, but, in our opinion, the concept of “chronic” is related to the duration of the infection, while “acute” is connected both to the duration and to the evident presence of the typical signs of inflammation.

Although probably important, these categories do not match, in our opinion, the concepts of High- and Low-Grade Infection. It is perfectly possible to have a chronic PJI, which lasts since many months or years, that presents itself as self-evident, with a draining fistula or an exposed implant and with all serum markers positive. This according to our definition is a “High-Grade infection”, since the effects of the infection can be easily detected by observation and by the available tests.

On the other side we may see patients complaining for unexplained pain two years after total knee replacement, with conflicting exams and borderline tests. Is this an acute or a chronic infection ? We should say it is chronic, since probably it dates back to the time of surgery. But it has peculiar aspects when compared to other “classical” chronic infection described above. This is a “Low-Grade Infection” or a malfunction of the implant, related to biofilm-producing bacteria, that trigger only a mild response in the host.

The reason for introducing the distinction between High- and Low-Grade Infection relies on the biofilm-related nature of implant-related infections, which in turn is the reason for the difficult diagnosis that we face in many cases.

This are the conditions that we think is necessary to differentiate in a modern definition of PJI, since these conditions present different diagnostic challenges, may need different cut-offs and may also have different indications and outcomes of treatments. .

The definition of "Biofilm-related Implant Malfunction (BIM)" is ambiguous. It seems to refer to a situation in which, although the clinical presentation includes the possible diagnosis of a chronic infection, the pre- and intra-operative are not conclusive, and the diagnosis of infection is achieved postoperatively. Therefore, this entity is finally a PJI, and not another different entity.

Once again, ours is, at the moment, a “concept paper”. However, a validation trial is ongoing in 5 different world center and the preliminary results do show that patients that meet the Biofilm-Implant Malfunction definition does exist. Maybe according to other definitions all these infection would have been missed. Are these cases difficult to diagnose ? for sure. But until we do not give them a name we cannot try to refine our diagnostic tools to identify them. And this does not necessarily mean to include new tests. We should consider that all available studies to validate a diagnostic test have been performed comparing well established infections with patients with well functioning implants. So also the cut-offs have been set to diagnose these “High-Grade infections” and this may explain why, using the same cut-offs, we have so many problem in identifying other less evident conditions.

Probably the inclusion of 3 concepts: non-infection, contamination and PJI would be enough. The “PJI category” would include the possible clinical presentations mentioned (chronic and acute); the pre-and intraoperative results could be less than 0, not suggesting in principle an infection, equal than 0, suggesting a possible infection, or more than 0, indicating a high suspicion of infection.

This is exactly what current definitions are trying to do. And they all fail…

Look at the last Consensus definition: Preoperative diagnosis, score 2-5 - Decision: “Possibly Infected”; Intra-operative diagnosis: score 4-5 – Decision: Inconclusive.

What is the information that this kind of definition is giving ? Our goal is to move a step forward and our tool is to identify intermediate conditions, that exist and that only were not defined before…

Do the cases included in the Table 4 correspond to real patients? It would be advisable to indicate it. Is surprising that the Case 2 has a synovial fluid WBC negative, but the leukocyte esterase is positive; given the correlation between the two markers, it should be infrequent.

The data are from real patients. Now made clear both in the text and in the Table. That particular patient had 1380 cells/μL.   

FORMAL ASPECTS

In the Introduction, line 81, it is mentioned the potential “conflict of interests” of the experts involved in the realization of previous PJI, definitions. I guess that refers to the inclusion of the alpha-defensin as a criterion for the ICM 2018 PJI definiton; beyond this criterion, I see no clear conflicts of interest in previous definitions. Such a general expression about “conflict of interests” raises doubts about all the previous definitions that do not seem to conform to reality. Therefore, this mention should be reconsidered or narrow down.

Now removed.

In the Introduction, Lines 85-87 it is said "while laboratory tests with similar sensitivity or specificity cannot be used alternatively, thus restricted the choice for physicians that operate in scarce resources environment and prompting the adoption of novel, often more expensive, biomarkers". Again, it seems to refer to the ICM 2018 definition. However, being fair, this definition includes as similar alternatives "elevations of synovial WBC count, leukocite esterase and alpha-Defensin"; of these 3 tests, only the last one is an expensive biomarker and is not included as essential. The rest of the PJI definitions don't includes expensive tests. So, these sentences should be rewritten.

To be fair, in the original definition, the one used in the validation study published in J Arthopl in 2018 (cf. Parvizi J, Tan TL, Goswami K, Higuera C, Della Valle C, Chen AF, Shohat N. The 2018 Definition of Periprosthetic Hip and Knee Infection: An Evidence-Based and Validated Criteria. J Arthroplasty. 2018 May;33(5):1309-1314.e2. doi:10.1016/j.arth.2018.02.078), alpha-defensin was scored 3 as a stand alone test, while the other two (LE and WBC) were set as a different criteria, also scored 3. Only after many and repeated solicitations by several experts around the world, finally alpha-defensin was included with the other two tests…

But ours is a more general message. In fact our proposed definition does not exclude expensive imaging techniques or markers, provided that they meet the minimum sensitivity and specificity requirements, but leaves the surgeon free to decide.

We think that striving the concept that, as a general rule, definitions should include all the tests with similar sensitivity and/or specificity, scoring them the same and then leaving the surgeons free to choose the most appropriate in a given patient, is pivotal.

It is hard to understand why we should assign 2 points to D-Dimer and 3 to elevated WBC. On which basis ? There is no scientific explanation.

Why alpha-defensin should be rated 3 and a positive cultural examination 2 ? Why if, for example according to IDSA, even a single positive test can make the diagnosis of PJI, if a virulent pathogen is isolated ?

What is the justification to give 1, 2 or 3 points ? we do not know… it is just a compromise, like a vote, like the approach of the Consensus. But science may not be decided by a vote, as Galileo taught to us some centuries ago…

It is not clear the relationship between the mentioned in lines 63-65 and the reference number 3.

Reference now removed.

Table 1: It would be advisable to improve and order the presentation of this table:

Similar criteria should be presented in the same row and expressed in a similar way to facilitate the comparison between definitions. The MSIS 2011 and ICM 2013 definitions practically only differ in the criterion of the periprosthetic purulence and yet they seem very different definitions at first sight

Sorry but we disagree on this point. The Criteria of 2011 and 2013 differ in various aspects, including:

a.     The number of minor criteria to be fulfilled to define a PJI (4 of 6 in 2011 and 3 of 5 in 2013)

b.    Leukocyte esterase is a criteria in 2013 but not in 2011

c.     Purulence is a criteria in 2011, but not in 2013

Given the heterogeneity of the different definitions, if we put each criteria in a row, we will end with a very long Table. Even if this would further strengthen our message, we think that this is practically no feasible. In any case, if the Editor agrees on that we may re-fomat the Table accordingly.

Whenever the definition includes cut-off of quantitative variables, these should be shown (in table or table-foot)

The ICM 2018 definition gives different cut-offs in the case of acute and chronic infections, which should be mentioned

Cut-offs have now been included in Table 1 for all definitions and for acute and chronic infections, according to ICM 2018.

The ICM 2018 definition do not refers to "criteria" but points or score

We disagree. In the original booklet, in response to Question 1. “What is the definition of a …” (cf. https://icmphilly.com/questions/what-is-the-definition-of-a-periprosthetic-joint-infection-pji-of-the-knee-and-the-hip-can-the-same-criteria-be-used-for-both-joints/ ) the Figure 1. is entitled “Proposed 2018 ICM Criteria for PJI.

The IDSA definition includes the possibility of a single culture with a virulent microorganism as a diagnostic criterion, which should be also mentioned

Now added in the Table 1.

Table 2: the title must correspond to "Pre-" or Intraoperative Tests (it seems that the "pre-" was left out by mistake.

Thank you for the comment. This is a typing error. Now corrected to: “Pre- and intra-operative tests”

Table 3: The acronyms used should be defined at the table footnote

Now added.

Line 130: The first time a microorganism is mentioned, the genus must be written without shortening (here, it would not be right "C. acnes”); on the other hand, currently Propionibacterium acnes is named Cutibacterium acnes. So, here it is recommended: Cutibacterium (previously named Propionibacterium) acnes.

Line 140: Here, however, it would be appropriate: "C. acnes".

Both amended, thank you..

 In the Discussion is properly mentioned the difficulty of validating the PJI definitions for various reasons. It is criticized, in this sense, the validation carried out of the 2018 ICM definition. It is certainly a definition and a validation that presents problematic aspects. However, it is not clear how the authors of the current definition seek to circumvent the difficulties associated with the validation of their definition and whether this validation will be better than the previously mentioned. Additionally, it should be avoided to make literal quotes of what other authors mention and reference it with an indirect style. These aspects should be taken into account in the discussion.

A large part of the quotes has now been removed, also according to the suggestions of Reviewer 1.

The sentence concerning future validation of our proposed definition has now been changed to:

Aware of the challenges posed by clinical validations, our future target is to first retrospectively apply the proposed WAIOT definition to a large population of patients, comparing the results with other existing definitions; as a second step a prospective multinational trial aimed at comparing pre-operative and intra-operative application of the definition with post-operative data is planned.”

  Round  2

Reviewer 3 Report

Thank you very much for your extensive review and answers to all the raised points.

This is a very interesting new aproximation to the diagnosis of PJI. I still have some doubts about the appropriateness of the given names to the different types of PJI (BIM, LG-PJI, HG-PJI), although I undestand the purpose of the authors. I feel that these three categories represents different presentations of infection and not necessarily "low or high grade" infection (a concept difficult to understand from an ID point of view). Anyway, I believe the most important is the underlying concepts and criteria (more tan the names of the categories).    

J. Clin. Med. EISSN 2077-0383 Published by MDPI AG, Basel, Switzerland RSS E-Mail Table of Contents Alert
Back to Top